# Hereditary Connective Tissue Diseases and Risk of Post-Acute SARS-CoV-2

**DOI:** 10.3390/v16030461

**Published:** 2024-03-17

**Authors:** Maggie L. Bartlett, Daniel Sova, Mahim Jain

**Affiliations:** 1W. Harry Feinstone Department of Molecular Microbiology and Immunology, Johns Hopkins Bloomberg School of Public Health, Baltimore, MD 212051, USA; 2John Hopkins Medicine, Physical Medicine and Rehabilitation, Baltimore, MD 212052, USA; 3Bone Disorders Program, Kennedy Krieger Institute, Baltimore, MD 21205, USA; jainm@kennedykrieger.org

**Keywords:** connective tissue disease, PASC

## Abstract

We completed a retrospective review of data collected by the JH-CROWN consortium based on ICD10 codes for a hospitalized cohort. The severity and prevalence of COVID-19 and development of PASC within heritable connective tissue diseases were unknown; however, clinical observation suggested a thorough examination was necessary. We compared rates of disease severity, death, and PASC in connective tissue diseases versus the entire cohort as well as in diabetes and hypertension to determine if connective tissue disease was a risk factor. Of the 15,676 patients in the database, 63 (0.40%) had a connective tissue disease, which is elevated relative to the distribution in the population, suggesting a higher risk of severe disease. Within these 63 patients, 9.52% developed PASC compared to 2.54% in the entire cohort (*p* < 0.005). Elucidation of populations at high risk for severe disease and development of PASC is integral to improving treatment approaches. Further, no other study to date has examined the risk in those with connective tissue diseases and these data support a need for enhanced awareness among physicians, patients, and the community.

## 1. Introduction

By December 2020, the pandemic caused by severe acute respiratory syndrome virus 2 (SARS-CoV-2) had claimed >6.5 million lives, and its effects are ongoing [1]. SARS-CoV-2 disease (COVID-19) is predominately characterized by hyperinflammation, pneumonia, respiratory failure, prothrombotic state, cardiac dysfunction, high mortality, and persistent symptoms in some, known as post-acute SARS-CoV-2 (PASC) [2]. While the delineation of PASC subtypes is still being defined, the WHO defines PASC as “…occurring in individuals with a history of probable or confirmed SARS-CoV-2 infection, usually 3 months from the onset of COVID-19 with symptoms and that last for at least 2 months and cannot be explained by an alternative diagnosis” [3]. While this is an evolving field of study, a substantial portion of patients have been reported to be affected, with numbers as high as 50% [2]. PASC may include, and is not limited to, cognitive defects, gastrointestinal distress, fatigue, brain fog, and shortness of breath. While PASC has been associated with acute disease, it has been documented regardless of severity and presents with wide heterogeneity among those affected [2]. PASC can be present in any demographic, including the pediatric population, and evidence to date has shown that women can be twice as likely to develop PASC as compared to men [4]. However, above the age of 60, there no longer seems to be a persistent gender discrepancy [5]. Post-viral diseases like PASC have been well-documented for many viral illnesses; however, acute febrile illness could not be directly correlated to long-term sequelae due to a lack of substantial awareness and high case counts [6,7,8,9]. Furthermore, few have focused on the predisposition to post-viral diseases within rare heritable diseases, particularly those that affect connective tissue.

Hereditary connective tissue diseases (hCTDs) or genetic collagen disorders include a broad grouping of syndromes that range from mild to debilitating. In this study, we focus on specific collagen disorders such as Ehlers–Danlos syndrome (EDS; types 3 and 5 collagen), Stickler Syndrome (STL; types 2, 9, and 11 collagen), and Osteogenic imperfecta (OI; type 1 collagen). Approximately 0.3% of people suffer from an hCTD, which is likely an underestimation due to lack of diagnostic criteria and awareness in the clinical setting outside of specialized clinics. The conditions involve multiple body systems, with common clinical features amongst these conditions including joint hypermobility, vascular abnormalities, risk of injuries/fractures, and risk of chronic pain and fatigue [10]. This population was hypothesized to have an increased propensity for development of post-viral diseases including PASC.

To discern the prevalence of COVID-19 severity and PASC among patients with hCTDs, we utilized the COVID-19 Precision Medicine Analytics Platform Registry (JH-CROWN) to conduct a retrospective analysis [11]. We found that individuals with hCTDs had higher rates of COVID-19 as well as PASC compared to in the general population as well as in well-known comorbidities that increase risk such as hypertension (HT) and diabetes (DB).

## 2. Materials and Methods

This represents a retrospective cohort analysis of patients who were tested and treated for SARS-CoV-2 between 11 March 2020 and 31 August 2022 at the Johns Hopkins Medicine healthcare system locations in Maryland. Design and inclusion criteria for admitted SARS-CoV-2 patients have been in part described elsewhere [12]. All patients were positive for SARS-CoV-2 nucleic acid. The JH-CROWN registry was utilized, which taps into the Johns Hopkins Precision Medicine Analytics Platform to extract health records, demographics, medical history, comorbidities, symptoms, vital signs, respiratory events, medications, and laboratory results [11]. Institutional review boards of hospitals approved for use in this study deemed it as having minimal risk and waived consent requirements. We stratified data by hCTD, age, and sex. We defined hCTDs based on the reported ICD10 code or the presence of a diagnosis in the patient’s medical history for all variations of EDS (n = 30), STL (n = 22), OI (n = 9), EDS + STL (n = 1), or OI + EDS + STL (n = 1). To parse development of PASC, ICD10 codes were used and fell within the following subgroups: EDS (n = 4), STL (n = 1), and OI (n = 1). We included analyses of rates of diabetes and hypertension, which have been established as risk factors for severe COVID-19 and PASC [13,14]. Primary outcomes were defined by using the World Health Organization (WHO) COVID-19 disease severity scale. This cohort of PASC patients represents 0.40% of the subjects in the CROWN database from which these data were collated and is likely an under-representation. A major hurdle of retrospective analyses like these is that they rely on the database to be capable of distinguishing phenotypes of a yet-to-be-well-defined syndrome. Given the complexities of these phenotypes, we preferred to focus only on the patients in which the ICD10 code for PASC were utilized. Of the 15,676 patients in the JH-CROWN database, those with diabetes (n = 4819), hypertension (n = 9402), hCTD (n = 63), and PASC (n = 398) were extracted by ICD10 code. For PASC, the database contained hCTD (n = 6), HT (n = 291), and DB (n = 163). Raw numbers and percentages for COVID-19 and PASC were compared using Chi-square tests to estimate differences by hCTD in R. Cohort characteristics were distinguished in R using Chi-square or Fisher’s exact tests while continuous variables were compared using *t*-tests or Wilcoxon rank sum tests, based on distribution type. Regression analyses were performed using the R package glm using logarithmic models and sequential addition of variables. While these data were extracted from a large pool of patients across hospitals in Maryland, the incidence of connective tissue diseases is rare in the general population. Further, we limited our analyses to those with reported diagnoses via the ICD10 code system which likely under-represents the population of interest.

## 3. Results

We were interested in the prevalence of hCTDs in PASC cohorts and their predisposition to long-term post-viral symptoms. During the SARS-CoV-2 pandemic, the JH-CROWN database was developed to catalog and document patient outcomes [11]. Clinical observation suggested that individuals with hCTDs were more likely to develop PASC; however, no retrospective analyses have been performed. Statistical analyses were computed on a data projection which included a diagnosis of hCTD (total n = 63) that could be further subgrouped based on the specific hCTD (EDS n = 30, OI n = 9, STL n = 22) or, in some cases, individuals with multiple hCTDs (OI + STL n = 1, OI + EDS + STL n = 1). Our dataset also included individuals without an hCTD and with a diagnosis of HT (n = 9402) or DB (n = 4819). Given the small number of individual hCTDs, we decided to group them to increase the power to identify their risk. Out of 15,676 individuals, 396 had the ICD10 code for PASC or one of the following reported diagnoses: long COVID-19 or post-acute COVID. Within this cohort, the percent of PASC was 2.54% vs. 9.52% within hCTDs (*p* < 0.005). Rates were compared to other known risk factors of severe disease such as HT (5.09%, *p* < 0.05) and DB (5.82%, *p* < 0.05) which, while higher than average within the cohort, were both lower than the rate for PASC in hCTDs.

### 3.1. Patients with hCTD Are at Higher Risk of/Predisposed to Hospitalization and Severe COVID-19 but Not Death

Compared to the rates in the general population, we found increased representation of hCTDs in the hospitalized cohort with a cumulative increase of ~4-fold overall compared to the higher end of predicted cumulative population rates (~0.11% cumulative hCTDs compared to 0.40% hCTDs in this cohort). Within the cohort, we compared the rates of hCTD, HT, and DB to population levels and found that hCTDs were also over-represented, similar to what has been reported for HT by Fisher’s exact testing (CTD vs. HT *p* > 0.05, DB vs. hCTD *p* > 0.05, HT vs. other *p* < 0.001, DB vs. other *p* < 0.001, hCTD vs. other *p* < 0.001). Our results were in line with previous analyses of the JH-CROWN cohort for rates of DB (30%) and HT (47%) [12]. These data suggest that the hCTD population is at higher risk and warrants awareness. We found that relative to patients with HT and severe vs. mild/moderate disease (based on WHO criteria), individuals with hCTDs were significantly more likely to have severe disease vs. mild/moderate (*p* < 0.05). Interestingly, while patients with hCTDs were over-represented in the COVID-19 hospitalized cohort, the odds ratio for death was not significant relative to death in the entire cohort (OR 1.50, *p* > 0.05, 95% CI 0.63 to 3.54) (Table 1). Of note, the two cases of death within 30 days of COVID-19 hospitalization were both in patients with STL, which is of interest for follow-up study. 

### 3.2. Patients with hCTD Do Not Have a Higher Risk of PASC Based on Variant or Vaccination

We were interested in whether there were variations by variants of concern and so performed comparisons based on likely variants circulating at the time of COVID-19 diagnosis. The comparisons revealed no biases in this cohort, although without sequencing data, a more granular analysis is of interest for follow-up study. We further examined the data to determine whether vaccination prevented PASC within hCTDs; however, as the majority of patients had developed PASC following primary infection in 2021 and received a vaccination later that year, further studies are anticipated to address this knowledge gap.

### 3.3. Patients with hCTD Are at Higher Risk of/Predisposed to PASC

We found that 9.52% of patients with an hCTD determined by ICD10 code subsequently received a diagnosis of PASC compared to 3.10% of patients with HT and 3.38% of patients with DB, which are leading conditions reported to date associated with developing PASC (Table 1). Relative to rates of PASC in the total cohort (2.54%), the rates in all three subpopulations were significantly higher based on the Fisher exact test: hCTD *p* < 0.005, HT *p*-value = 0.013, and DB *p*-value = 0.018, suggesting that people with hCTDs are also at increased risk for developing PASC. We were interested in whether individuals with hCTDs developed PASC sooner than those with HT or unknown diagnoses; however, we found no statistically significant differences with the average time from hospitalization to PASC diagnosis in those with hCTDs being 499.3 days versus 354.8 or 463.0 for HT or unknown, respectively. Odds ratios were computed between hCTD + PASC and HT + PASC, DB + PASC, or total + PASC which suggested that individuals with hCTDs were more than three times more likely than those with HT to develop PASC and more than four times more likely than the overall population; at the same time, there was a low association between PASC and HT or DB relative to the overall prevalence (Table 2). Using logarithmic regression, we assessed the role of comorbidities and confounding variables on model fit. hCTD had a significant impact, and the model was improved by the inclusion of HT, DB, or sex.

## 4. Discussion

Utilizing the JH-CROWN database, we compared rates of PASC in individuals with hCTDs like EDS, STL, and OI to those in individuals with HT and DB as well as estimated rates in the population. This study highlights the risk of hospitalization due to COVID-19 and indicates that the development of PASC in individuals with hCTDs is equivalent to or greater than in individuals with other known risk factors such as DB and HT [13,14]. We suspect that the elevated risk of PASC is due in part to the pathophysiology of hCTDs, which is exacerbated by viral disease, and that this association was likely underappreciated due to the rarity of hCTDs; however, more in depth mechanistic work is paramount to elucidating this. While there was not a higher likelihood of death among those with hCTDs and COVID-19, 5/7 individuals with hCTDs that died had STL, which is of interest for further investigation.

While most people recover from COVID-19 within days, the persistence of symptoms after acute illness remains a global health issue; however, the factors which predispose individuals to post-viral syndromes like PASC are poorly defined. Intriguingly, others have found that ~23% of PASC patients are thought to fit within the sub-phenotype of musculoskeletal/nervous system malfunction, which includes heritable hCTDs [15]. The pathophysiology of this risk is not well understood and requires further evaluation. Some potential associations are as follows: PASC shares hallmarks of other autonomic pathologies that are commonly associated with hCTDs including postural orthostatic tachycardia syndrome (POTS) and myalgic encephalomyelitis/chronic fatigue syndrome (ME/CFS) [12,15,16,17]. While the propensity to develop autonomic dysfunction post-acute viral illness is not limited to SARS-CoV-2 or those with hCTDs, what contributes to these outcomes is not well understood [15]. It is possible that viral infection exacerbates already compromised systems in hCTD patients’ cells which results in an altered propensity to develop PASC or other post-viral diseases; however, more research is required to reveal potential mechanisms. 

Patients with hCTDs are a heterogeneous population with an average time to diagnosis of 10–12 years in part due to multi-systemic symptoms. The European Reference Network on Rare and Complex Connective Tissue and Musculoskeletal Diseases (ERN ReCONNET) detailed the impacts on patients with rare connective tissue diseases (rCTD) and their communities which included reduced access to care, limitations of certain drugs, and disease relapses, which may be further exacerbated in these patients given these new findings [16]. While others have reported higher odds ratios for DB or HT and PASC, others have shown nominal differences, as we observed, which may be indicative of the strict inclusion criteria as well as differences in geographic factors [12]. Increasing understanding of the risks associated with hCTDs such as viral disease severity and post-viral diseases can aid in reduced morbidity and reduced mortality among an already challenging cohort of patients. These data support the link between hCTDs, COVID-19, and PASC; however, more research is needed to understand the molecular pathogenesis. Indeed, common comorbidities in those with hCTDs including postural orthostatic tachycardia syndrome (POTS), MCAS, and CE/MFS, may represent clinical confounds. Patients with a diagnosis of POTS made up 2.65% of patients, which is significantly greater than the estimated ~1% prevalence worldwide. None of the patients with hCTD and PASC were diagnosed with POTS, MCAS, or CE/MFS at admittance to the hospital. While these data were extracted from a large pool of patients across hospitals in Maryland, the incidence of collagen disorders and these comorbidities is rare in the general population. Further, we limited our analyses to those with reported diagnoses via the ICD10 code system, which likely under-represents the population of interest. Finally, as the spectrum of post-viral illness is vast and phenotypes are still being elucidated, it is not feasible currently to correlate specific outcomes beyond PASC as experts have yet to determine appropriate diagnostic criteria for the clinical setting. Understanding the cascade of events that leads to symptoms in these complex diseases is paramount to developing mitigation strategies. The somatic nervous system (SNS) and autonomic nervous system (ANS) make up the peripheral nervous system (PNS). While the SNS is important for collecting sensory information and facilitating voluntary functions, the ANS regulates unconscious actions. Damage to any portion of the nervous system during viral infection can result in long-lasting sequelae. Prospective studies have illuminated the similarities between some PASC subsets and POTS, which supports the potential for this to be in part a viral-induced/-triggered phenotype in hCTDs [17]. Different hCTDs have been associated with peripheral neuropathies ranging in prevalence from 10 to 60% [18]. The primary components suspected to be responsible are nerve lesions due to vasculitis or immune abnormalities, and typically, patients have a spectrum of overlapping possible contributing phenotypes. How these are triggered is unknown and thought to be a combination of genetics and environment. The propensity to develop autonomic dysfunction post-acute viral illnesses is not limited to SARS-CoV-2 [19,20,21,22,23]. Due to the underlying malfunction of key pathways, we hypothesize that viral infection exacerbates compromised systems in hCTD patients’ cells which results in the development of PASC (or other post-acute infection syndromes) and is of high interest for future studies. To our knowledge, this represents the first compilation of data related to the risk of SARS-CoV-2-induced syndromes in individuals with hCTDs.

## Figures and Tables

**Table 1 viruses-16-00461-t001:** Odds ratios for PASC between hCTD, HT, and DB.

Comparison	Odds Ratio	95% CI	Z	Significance
HT vs. hCTD death	2.76	1.15 to 6.61	2.27	*p* = 0.023
HT vs. DB death	1.74	1.13 to 2.69	2.50	*p* = 0.013
DB vs. hCTD death	1.58	0.63 to 3.96	0.98	ns
CTD vs. CROWN death	1.50	0.63 to 3.54	0.92	ns

**Table 2 viruses-16-00461-t002:** Odds ratios for PASC between hCTD, HT, and DB.

Comparison	Odds Ratio	95% CI	Z	Significance
hCTD PASC vs. total PASC	4.15	1.78 to 9.67	3.30	*p* = 0.0010
HT PASC vs. total PASC	1.23	1.05 to 1.43	2.62	*p* = 0.0089
DB PASC vs. total PASC	1.35	1.12 to 1.62	3.14	*p* = 0.0017
hCTD PASC vs. HT PASC	3.30	1.41 to 7.71	2.75	*p* = 0.0059
hCTD PASC vs. DB PASC	3.01	1.28 to 7.07	2.52	*p* = 0.012

## Data Availability

Data were sourced from the JH-CROWN database.

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
