# Peer review of "Hereditary Connective Tissue Diseases and Risk of Post-Acute SARS-CoV-2"

_viruses, 2024, doi:10.3390/v16030461_

Round 1

Reviewer 1 Report

Comments and Suggestions for Authors

I consider that the topic is relevant, important and new in this field. Materials, methods and results were well described. The conclusion is consistent. The references are appropriate. I have no additional comments on the tables.

Author Response

We thank the reviewer for their time and thoughtful comments.

Reviewer 2 Report

Comments and Suggestions for Authors

I enjoyed reading the article by Bartlett et al. The paper is well written and the objective is clearly defined. The results are interesting and nicely presented and their interpretation appears reasonable. A large database was utilized to provide information on very rare conditions such as patients with collagen disorders. Mechanistic approaches were not discussed, but this far beyond the scope of the study. 

I only have one major comment:

Please provide the characteristics of patients with CTD who developed PASC versus those with CTD who did not. Concomitant presence of hypertension and diabetes is particularly important based on the findings of this and other studies.

Other remarks:

- I suggest that the term "connective tissue diseases" changes to "collagen disorders" all throughout the text including the title. This is more accurate as connective diseases include a broad group of disorders, most commonly autoimmune rheumatic disordes, which however were not analyzed in the present report. Hence, the term "connective tissue diseases" appers rather misinformative.

- Please provide information on the statistical software that was used for statistical analysis. 

Author Response

We thank the reviewer for their feedback and have included supplemental table 1 to address the concomitant presence of HT/DB. While there are more in the PASC group with hCTDs and comorbidity, the regression modeling suggested only mild contribution from those covariates compared to the addition of CTDs. We have also altered the text on lines 47-48 to refer to these conditions as Hereditary connective tissue diseases (hCTDs) which are best described as collagen disorders.

Supplemental table 1:

Total

DB

HT

Black

White

Other

Female

Male

Age mean (range)

PASC

6

4

4

2

4

0

5

1

42.4 (23-69)

no PASC

57

27

27

11

45

7

45

18

47.3 (18-80)

Reviewer 3 Report

Comments and Suggestions for Authors

In this retrospective study, Bartlett et al aimed to investigate whether connective tissue diseases (CTD) increase the risk of post-acute COVID-19 syndrome (PASC).

This manuscript has several issues that should be underlined:

1. Methodology is quite confuse. Is it a case-control study (in this case it should be specified the populations of interest) or a retrospective cohort study (in this case exposures should be better specified)?

2. Results. Some passages are very difficult to interpret (e.g. lines 108-113). Full results of statistical analysis are missing. Table 1 with baseline characteristics is also missing.

3. The term CTD includes several diseases whose pathogenesis is very different. The Authors only focused on genetic ones. The term "Hereditary CTDs" should be preferred in this case.

4. Acknowledgment of limitations is missing in the Discussion. Apart from the ones related to the type of study, I would like to underline that EDS is known to be associated to POTS and MCAS, which are two very likely cofounders given that they can present clinical manifestations similar to PASC.

Comments on the Quality of English Language

No issues

Author Response

  1. Methodology is quite confuse. Is it a case-control study (in this case it should be specified the populations of interest) or a retrospective cohort study (in this case exposures should be better specified)? Retrospective, as noted on lines 67 in materials and methods and added again to line 61 in the intro for clarity. Additionally, we added more detail on lines 70-71 in which all admitted cases were PCR positive.
  2. Results. Some passages are very difficult to interpret (e.g. lines 108-113). Full results of statistical analysis are missing. Table 1 with baseline characteristics is also missing. We have revised lines 108-113 (now 108-111) for clarity. We appreciate the insight and have added it as supplementary table 1.

Total

DB

HT

Black

White

Other

Female

Male

Age mean (range)

PASC

6

4

4

2

4

0

5

1

42.4 (23-69)

no PASC

57

27

27

11

45

7

45

18

47.3 (18-80)

  1. The term CTD includes several diseases whose pathogenesis is very different. The Authors only focused on genetic ones. The term "Hereditary CTDs" should be preferred in this case. Revised, thank you for your input.
  2. Acknowledgment of limitations is missing in the Discussion. Apart from the ones related to the type of study, I would like to underline that EDS is known to be associated to POTS and MCAS, which are two very likely cofounders given that they can present clinical manifestations similar to PASC. Yes agree and appreciate the recommendation. Added lines 204-206 and a limitations section 209-217 with additional discussion.

Round 2

Reviewer 2 Report

Comments and Suggestions for Authors

The revised manuscript is significantly improved. 

I only have one further comment.

The authors use both the term CTDs and hCTDs (connective tissue diseases and hereditary connective tissue diseases) unequivocally throughout the text, which may create confusion to the reader. Please consider sonsistently using the term hCTDs both in the title/abstract and the main manuscript, as this is more appropriate. 

Author Response

Thank you for noting, we have revised to be consistent throughout. 

Reviewer 3 Report

Comments and Suggestions for Authors

The Authors addressed the majority of the comments.

I would suggest to add "Heritable" in the title, and use the acronym hCTDs for consistency across all the manuscript to distinguish from the autoimmune counterpart. 

Author Response

Thank you for suggesting, revised throughout.